# OPTIMIZING THE LATENT SPACE OF GENERATIVE NETWORKS

## ABSTRACT

Generative Adversarial Networks (GANs) have achieved remarkable results in the task of generating realistic natural images. In most applications, GAN models share two aspects in common. On the one hand, GANs training involves solving a challenging saddle point optimization problem, interpreted as an adversarial game between a generator and a discriminator functions. On the other hand, the generator and the discriminator are parametrized in terms of deep convolutional neural networks. The goal of this paper is to disentangle the contribution of these two factors to the success of GANs.

In particular, we introduce *Generative Latent Optimization* (GLO), a framework to train deep convolutional generators without using discriminators, thus avoiding the instability of adversarial optimization problems. Throughout a variety of experiments, we show that GLO enjoys many of the desirable properties of GANs: learning from large data, synthesizing visually-appealing samples, interpolating meaningfully between samples, and performing linear arithmetic with noise vectors.

## 1 INTRODUCTION

Generative Adversarial Networks (GANs) (Goodfellow et al., 2014) are a powerful framework to learn generative models of natural images. GANs learn these generative models by setting up an adversarial game between two learning machines. On the one hand, a generator plays to transform noise vectors into fake samples, which resemble real samples drawn from a distribution of natural images. On the other hand, a discriminator plays to distinguish between real and fake samples. During training, the generator and the discriminator learn in turns. First, the discriminator learns to assign high scores to real samples, and low scores to fake samples. Then, the generator learns to increase the scores of fake samples, as to fool the discriminator. After proper training, the generator is able to produce realistic natural images from noise vectors.

Recently, GANs have been used to produce high-quality images resembling handwritten digits, human faces, and house interiors (Radford et al., 2015). Furthermore, GANs exhibit three strong signs of generalization. First, the generator translates *linear interpolations in the noise space* into *semantic interpolations in the image space*. In other words, a linear interpolation in the noise space will generate a smooth interpolation of visually-appealing images. Second, the generator allows *linear arithmetic in the noise space*. Similarly to word embeddings (Mikolov et al., 2013), linear arithmetic indicates that the generator organizes the noise space to disentangle the nonlinear factors of variation of natural images into linear statistics. Third, the generator is able to to synthesize new images that resemble those of the data distribution. This allows for applications such as image in-painting (Iizuka et al., 2017) and super-resolution (Ledig et al., 2016).

Despite their success, training and evaluating GANs is notoriously difficult. The adversarial optimization problem implemented by GANs is sensitive to random initialization, architectural choices, and hyper-parameter settings. In many cases, a fair amount of human care is necessary to find the correct configuration to train a GAN in a particular dataset. It is common to observe generators with similar architectures and hyper-parameters to exhibit dramatically different behaviors. Even when properly trained, the resulting generator may synthesize samples that resemble only a few localized regions (or modes) of the data distribution (Goodfellow, 2017). While several advances have been made to stabilize the training of GANs (Salimans et al., 2016), this task remains more art than science.

The difficulty of training GANs is aggravated by the challenges in their evaluation: since evaluating the likelihood of a GAN with respect to the data is an intractable problem, the current gold standard to evaluate the quality of GANs is to eyeball the samples produced by the generator. The evaluation of discriminators is also difficult, since their visual features do not always transfer well to supervised tasks (Donahue et al., 2016; Dumoulin et al., 2016). Finally, the application of GANs to non-image data has been relatively limited.

**Research question** To model natural images with GANs, the generator and discriminator are commonly parametrized as deep Convolutional Networks (convnets) (LeCun et al., 1998). Therefore, it is reasonable to hypothesize that the reasons for the success of GANs in modeling natural images come from two complementary sources:

(A1) Leveraging the powerful inductive bias of deep convnets.

(A2) The adversarial training protocol.

This work attempts to disentangle the factors of success (A1) and (A2) in GAN models. Specifically, we propose and study one algorithm that relies on (A1) and avoids (A2), but still obtains competitive results when compared to a GAN.

**Contribution.** We investigate the importance of the inductive bias of convnets by removing the adversarial training protocol of GANs (Section 2). Our approach, called *Generative Latent Optimization* (GLO), maps one *learnable* noise vector to each of the images in our dataset by minimizing a simple reconstruction loss. Since we are predicting *images from learnable noise*, GLO borrows inspiration from recent methods to predict *learnable noise from images* (Bojanowski and Joulin, 2017). Alternatively, one can understand GLO as an auto-encoder where the latent representation is not produced by a parametric encoder, but learned freely in a non-parametric manner. In contrast to GANs, we track of the correspondence between each learned noise vector and the image that it represents. Hence, the goal of GLO is to find a meaningful organization of the noise vectors, such that they can be mapped to their target images. To turn GLO into a generative model, we observe that it suffices to learn a simple probability distribution on the learned noise vectors.

In our experiments (Section 3), we show that GLO inherits many of the appealing features of GANs, while enjoying a much simpler training protocol. In particular, we study the efficacy of GLO to compress and decompress a dataset of images (Section 3.3.1), generate new samples (Section 3.3.2), perform linear interpolations and extrapolations in the noise space (Section 3.3.3), and perform linear arithmetics (Section 3.3.5). Our experiments provide quantitative and qualitative comparisons to Principal Component Analysis (PCA), Variational Autoencoders (VAE) and GANs. We focus on the CelebA and LSUN-Bedroom datasets. We conclude our exposition in Section 5.

## 2 THE GENERATIVE LATENT OPTIMIZATION (GLO) MODEL

First, we consider a large set of images $\{x_1, \ldots, x_N\}$, where each image $x_i \in \mathcal{X}$ has dimensions $3 \times w \times h$. Second, we initialize a set of $d$-dimensional random vectors $\{z_1, \ldots, z_N\}$, where $z_i \in \mathcal{Z} \subseteq \mathbb{R}^d$ for all $i = 1, \ldots N$. Third, we pair the dataset of images with the random vectors, obtaining the dataset $\{(z_1, x_1), \ldots, (z_N, x_N)\}$. Finally, we jointly learn the parameters $\theta$ in $\Theta$ of a generator $g_\theta : \mathcal{Z} \to \mathcal{X}$ and the optimal noise vector $z_i$ for each image $x_i$, by solving:

$$\min_{\theta \in \Theta} \ \frac{1}{N} \sum_{i=1}^{N} \left[ \min_{z_i \in \mathcal{Z}} \ \ell\left(g_\theta(z_i), x_i\right) \right], \tag{1}$$

In the previous, $\ell : \mathcal{X} \times \mathcal{X}$ is a loss function measuring the reconstruction error from $g(z_i)$ to $x_i$. We call this model Generative Latent Optimization (GLO).

**Learnable $z_i$.** In contrast to autoencoders (Bourlard and Kamp, 1988), which assume a parametric model $f : \mathcal{X} \to \mathcal{Z}$, usually referred to as the *encoder*, to compute the vector $z$ from samples $x$, and minimize the reconstruction loss $\ell(g(f(x)), x)$, in GLO we jointly optimize the inputs $z_1, \ldots, z_N$ and the model parameter $\theta$. Since the vector $z$ is a free parameter, our model can recover all the solutions that could be found by an autoencoder, and reach some others. In a nutshell, GLO can be viewed as an "encoder-less" autoencoder, or as a "discriminator-less" GAN.

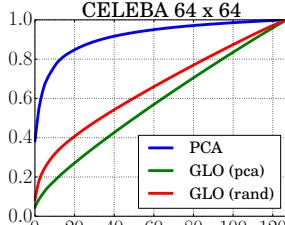 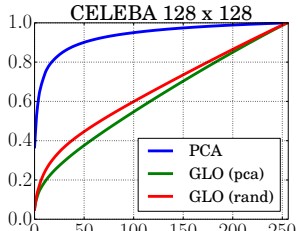 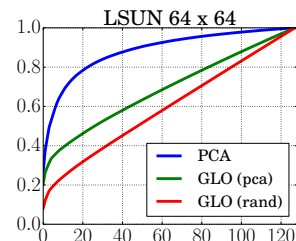

Figure 1: Plot of the cumulative sum of the singular values of the optimal $Z^*$ matrix. We observe that the proposed GLO model has a better conditioned covariance matrix and therefore better fills the latent space.

**Choice of $\mathcal{Z}$.** The representation space $\mathcal{Z}$ should encapsulate all of our *prior knowledge* about the data $\{x_1, \ldots, x_N\}$. Since we are interested in matching the properties of GANs, we make similar choices to them when it comes to the representation space $\mathcal{Z}$. The most common choices of the representation space for GANs are either the uniform distribution on the hypercube $[-1, +1]^d$, or the Normal distribution on $\mathbb{R}^d$. In previous literature, Gaussian distributions lead to more stable GAN training (Radford et al., 2015), we will take this choice to design our representation space. In GLO, the random vectors $z$ are learnable and can therefore attain any value during the training process. To avoid extremely large values, we normalize our learnable noise vectors $z$ at all times, to lay on the unit $\ell_2$ sphere.

**Choice of loss function.** On the one hand, the squared-loss function $\ell_2(x, x') = \|x - x'\|_2^2$ is a simple choice, but leads to blurry (average) reconstructions of natural images. On the other hand, GANs use a convnet (the discriminator) as loss function. Since the early layers of convnets focus on edges, the samples from a GAN are sharper. Therefore, our experiments provide quantitative and qualitative comparisons between the $\ell_2$ loss and the Laplacian pyramid $\text{Lap}_1$ loss

$$\text{Lap}_1(x, x') = \sum_j 2^{2j} |L^j(x) - L^j(x')|_1,$$

where $L^j(x)$ is the $j$-th level of the Laplacian pyramid representation of $x$ (Ling and Okada, 2006). Therefore, the $\text{Lap}_1$ loss weights the details at fine scales more heavily. In order to low-frequency content such as color information, we will use a weighted combination of the $\text{Lap}_1$ and the $\ell_2$ costs.

**Optimization.** For any choice of differentiable generator, the objective (1) is differentiable with respect to $z$, and $\theta$. Therefore, we will learn $z$ and $\theta$ by Stochastic Gradient Descent (SGD). The gradient of (1) with respect to $z$ can be obtained by backpropagating the gradients through the generator function (Bora et al., 2017). We project each $z$ back to the representation space $\mathcal{Z}$ after each update. To have noise vectors laying on the unit $\ell_2$ sphere, we project $z$ after each update by dividing its value by $\max(\|z\|_2, 1)$. We initialize the $z$ by either sampling them from a gaussian distribution or by taking the whitened PCA of the raw image pixels.

## 3 EXPERIMENTS

We organized our experiments as follows. First, Section 3.1 describes the generative models that we compare, along with their implementation details. Section 3.2 reviews the image datasets used in our experiments. Section 3.3 discusses the results of our experiments, including the compression of datasets (Section 3.3.1), the generation (Section 3.3.2) and interpolation (Section 3.3.3) of samples, and the results of arithmetic operations with noise vectors (Section 3.3.5).

### 3.1 METHODS

Throughout our experiments, we compare three different models with a representation space (noise vectors) of $d = 256$ dimensions.

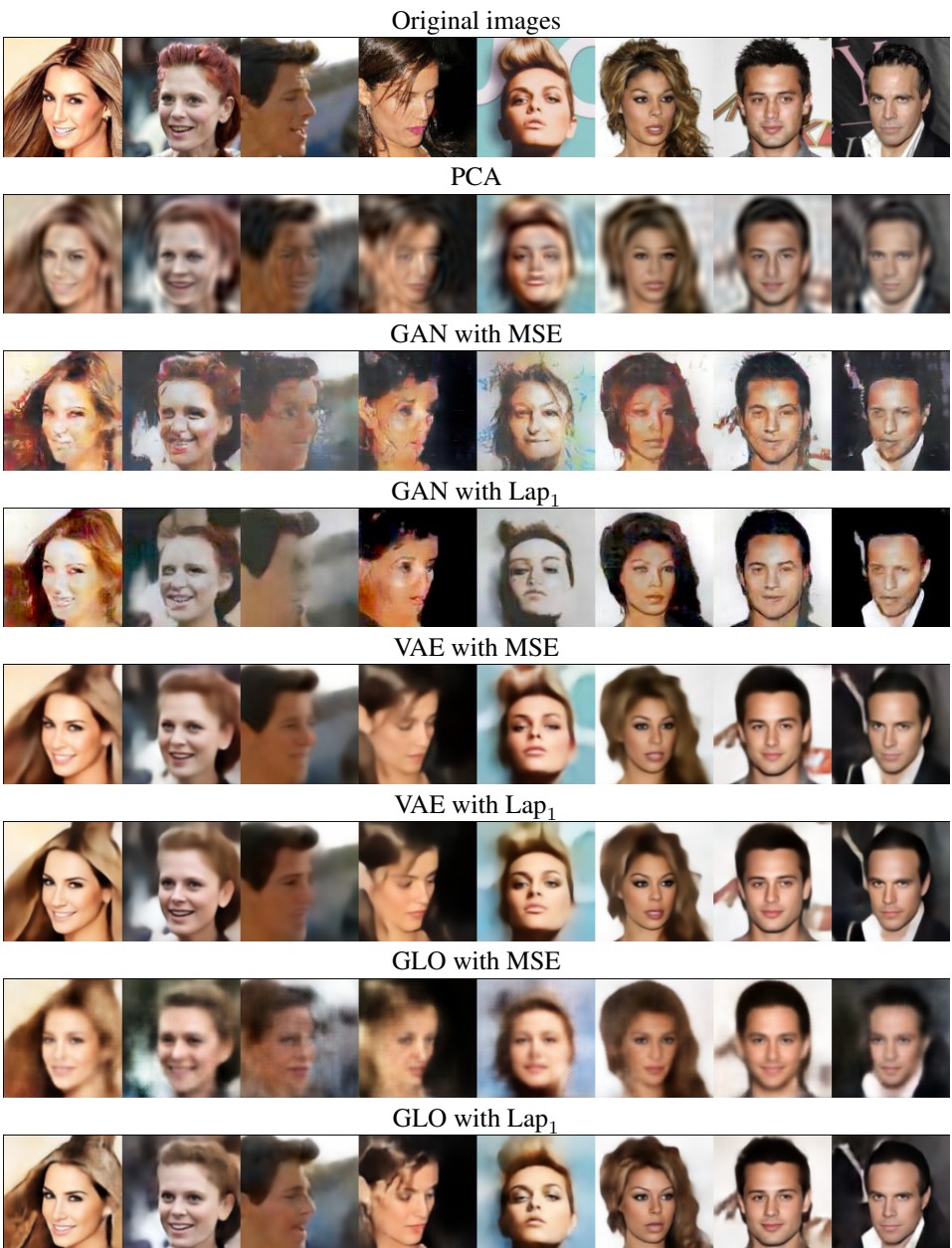

Figure 2: Reconstruction of training examples from the CelebA $128 \times 128$ dataset.

First, **PCA** (Pearson, 1901), equivalent to a linear autoencoder (Baldi and Hornik, 1989), where we retain the top 256 principal components.

Second, **DCGAN** (Radford et al., 2015). Since GANs do not come with a mechanism (inverse generator) to retrieve the random vector $g^{-1}(x)$ associated with an image $x$, we estimate this random vector by 1) instantiating a random vector $z_0$, and 2) computing updates $z_{i+1} = z_{i+1} - \alpha \frac{\partial \ell(g(z_i), x)}{\partial z_i}$ by backpropagation until convergence, where $\ell$ is either the $\ell_2$ or the $\text{Lap}_1$ loss. Our experiments measuring reconstruction error are disadvantageous to GANs, since these are models that are *not* trained to minimize this metric. We use the Adam optimizer (Kingma and Ba, 2015) with the parameters from (Radford et al., 2015).

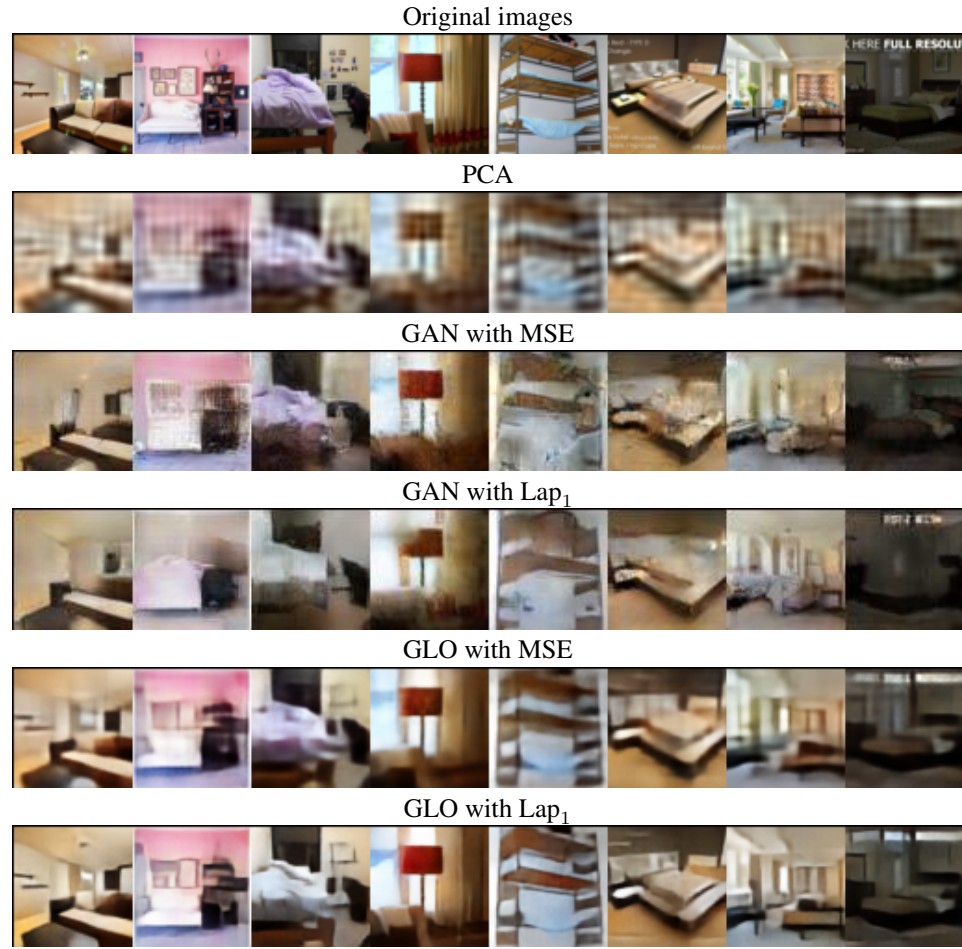

Figure 3: Reconstruction of training examples from the LSUN 64×64 dataset.

| Method | CelebA-64 | CelebA-128 | LSUN-64 |
|---|---|---|---|
| PCA | 0.0203 | 0.0132 | 0.0269 |
| GAN, MSE | 0.0255 | 0.0264 | 0.0262 |
| GAN, Lap$_1$ | 0.0399 | 0.0400 | 0.0403 |
| VAE, MSE | | 0.0122 | |
| VAE, Lap$_1$ | | 0.0147 | |
| GLO, MSE, random init. | 0.0326 | 0.0345 | 0.0957 |
| GLO, MSE, PCA init. | 0.0148 | 0.0142 | 0.0240 |
| GLO, Lap$_1$, random init. | 0.0175 | 0.0152 | 0.0444 |
| GLO, Lap$_1$, PCA init | 0.0130 | 0.0125 | 0.0222 |

Table 1: Reconstruction errors in MSE. We consider methods using both MSE and Lap$_1$ loss. We also specify the initialization method between random and PCA.

Third, **VAE** (Kingma and Welling, 2013). We train a VAE with the same encoder and decoder architectures as DCGAN. We train it with the default hyper-parameters for 25 epochs.

Third, **GLO (proposed model)**. We will train a GLO model where the generator follows the same architecture as the generator in DCGAN. We use Stochastic Gradient Descent (SGD) to optimize both $\theta$ and $z$, setting the learning rate for $\theta$ at 1 and the learning rate of $z$ at 10. After each uddpate, the noise vectors $z$ are projected to the unit $\ell_2$ sphere. In the sequel, we initialize the random vectors

of GLO using a Gaussian distribution (for the CelebA dataset) or the top $d$ principal components (for the LSUN dataset).

## 3.2 DATASETS

We evaluate all models on two datasets of natural images. Unless specified otherwise, we use the prescribed training splits to train our generative models. All the images are rescaled to have three channels, center-cropped, and normalized to pixel values in $[-1, +1]$.

First, **CelebA** (Liu et al., 2015) is a set of $202,599$ portraits of celebrities. We use the *aligned and cropped* version, scaled to $128 \times 128$ pixels.

Second, **LSUN** (Xiao et al., 2010) is a set of millions of images of scenes belonging to different scene categories. Following the tradition in GAN papers (Radford et al., 2015), we use the $3,033,042$ images belonging to the *bedroom* category. We resize the images to $64 \times 64$ pixels.

## 3.3 RESULTS

We compare the methods described on Section 3.1 when applied to the datasets described on Section 3.2. In particular, we evaluate the performance of the methods in the tasks of compressing a dataset, generating new samples, performing sample interpolation, and doing sample arithmetic.

### 3.3.1 DATASET COMPRESSION

We start by measuring the reconstruction error in terms of the mean-squared loss $\ell_2(x, x') = \|x - x'\|_2^2$ and the $\mathrm{Lap}_1$ loss (2) Table 1 shows the reconstruction error of all models and datasets for the $\ell_2$. This gives a rough idea about the coverage of each model over the dataset.

Figure 3 and 2 shows a few reconstruction examples obtained with fixed size latent space of various models. Figure 1 show the quantity of the representation space explained as a function of the number of eigenvectors used to reconstruct it. GLO trained from a random initialization is more aggressive about using the full representation space to spread the information around while PCA or autoencoders tend to concentrate the information in a few directions.

For completeness, we computed image reconstructions for the various models on a held-out set of images. To this end we use face images from deep funneled images from Labeled Faces in the Wild (Huang et al., 2012). In order to make the images similar to those found in CelebA we crop the images so as to align the location of eyes. The reconstructions of a random sample of images are presented in Fig. 10.

### 3.3.2 GENERATION OF NEW SAMPLES

Figure 4 shows samples from the each of the models on the CelebA dataset, and Figure 5 shows the same fro the LSUN dataset. In the case of GLO, we fit a Gaussian distribution with full covariance to the representation space $\mathcal{Z}$, and sample from such distribution to generate new samples. We can see that the samples are visually appealing even when placing such a simple probabilistic model on the representation space. We leave more careful modeling of $\mathcal{Z}$ for future work.

### 3.3.3 SAMPLE INTERPOLATION

Figures 6 and 7 show interpolations between different reconstructed training examples from the CelebA and LSUN datasets. We compare interpolations in $z$-space (where we linearly interpolate between two noise vectors and forward them to the model), linear interpolation in image space, interpolation in principal components, and interpolation in GAN $z$ space (where the endpoints to reconstruct training examples are obtained by optimization). The interpolations in $z$-space are very different from the interpolations in image space, showing that GLO learns a non-linear mapping between noise vectors and images.

GAN

VAE with Lap$_1$

GLO with Lap$_1$

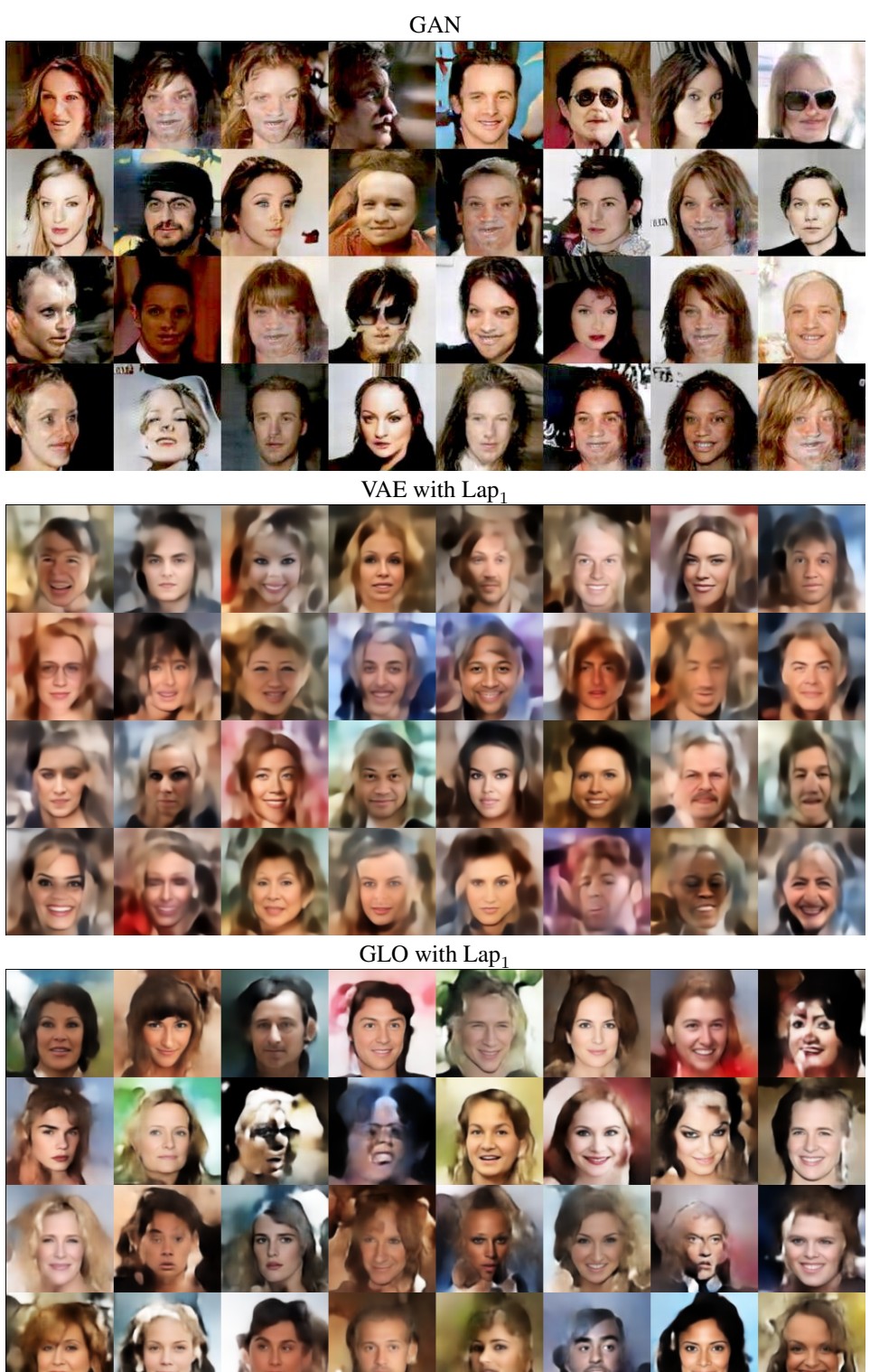

Figure 4: Generation of samples on the CelebA $128\times128$ dataset.

GAN

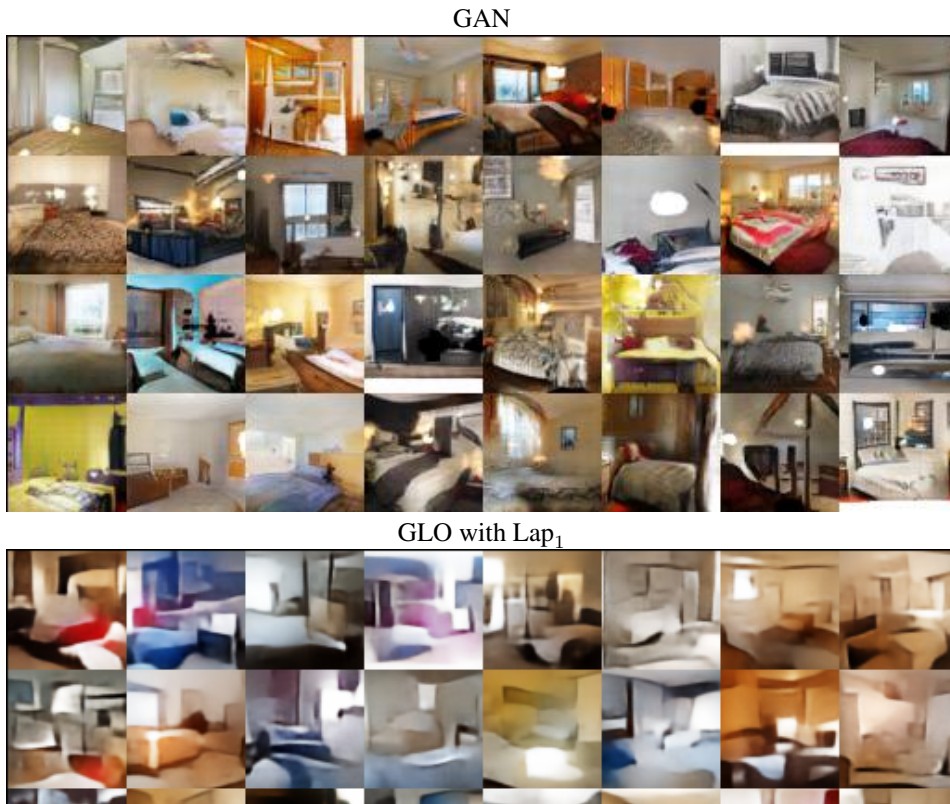

GLO with Lap$_1$

Figure 5: Generation of samples on the LSUN 64×64 dataset.

### 3.3.4 INTERPRETABILITY OF THE LATENT SPACE

The latent space can be explored by decomposing the covariance matrix of the latent vectors and moving along the eigenvectors associated with the largest eigenvalues from an image. The resulting image transformation often contains information about attributes that varies in the dataset. Figure 8 show some examples of image deformation along the principal axes. The image in the middle is the original image. Moving in either direction along an axis produces the images on its left and its right. We see that the main axes seem to contain information about standard face attributes. For example, the 4th component seems to be capturing information about facial expression while the 9th one seems to be capturing information about the age. In absence of supervision, some directions make several attributes move simultaneously, for example smiling seems correlated with the hair color. These correlations are artifacts of the CelebA dataset distribution.

### 3.3.5 NOISE VECTOR ARITHMETIC

In the spirit of Radford et al. (2015), we showcase the effect of simple arithmetic operations in the noise space of the various models. More precisely, we average the noise vector of three images of men wearing sunglasses, remove the average noise vector of three images of men not wearing sunglasses, and add the average noise vector of three images of women not wearing sunglasses. The resulting image resembles a woman wearing sunglasses glasses, as shown in Figure 9.

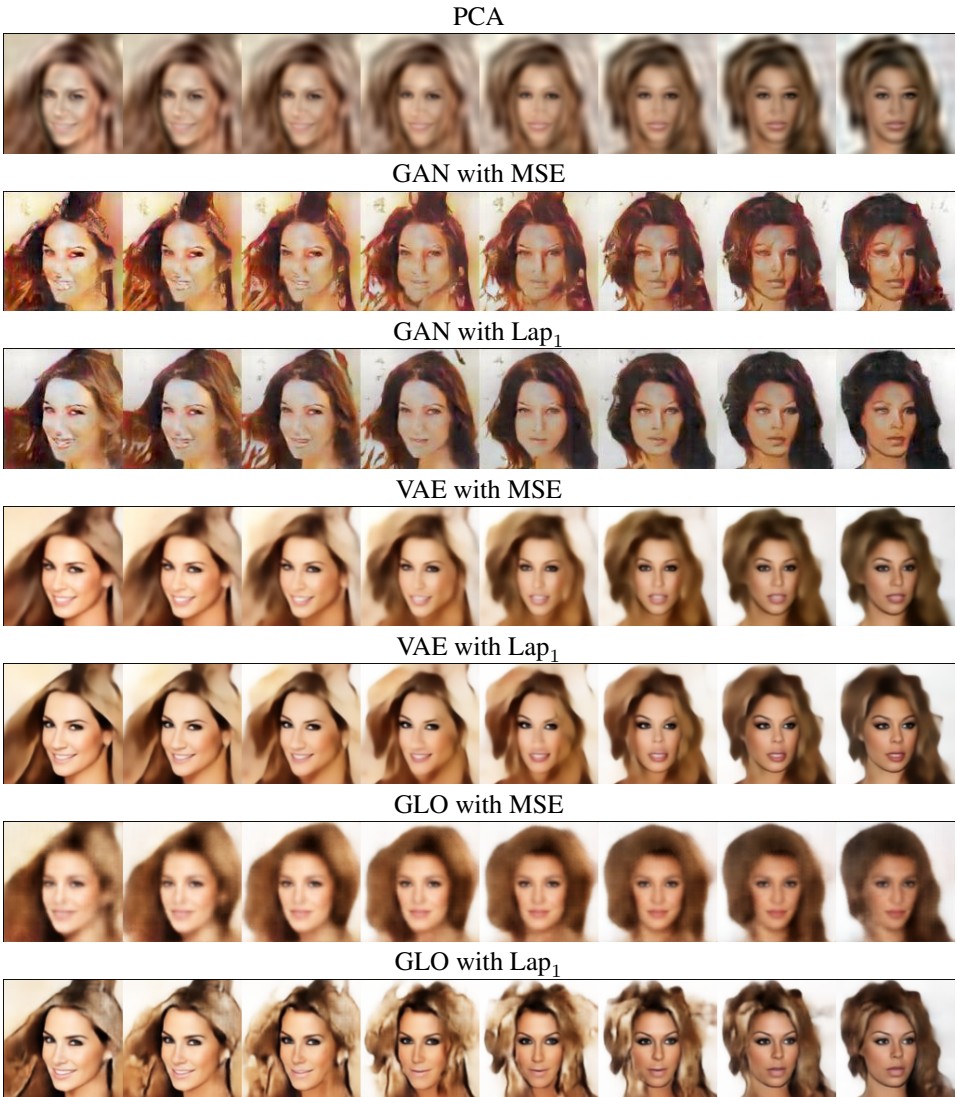

Figure 6: Interpolation of training examples on the CelebA 128×128 dataset.

## 4 RELATED WORK

**Generative Adversarial Networks.** GANs were introduced by Goodfellow et al. (2014), and refined in multiple recent works (Denton et al., 2015; Radford et al., 2015; Zhao et al., 2016; Salimans et al., 2016). As described in Section 1, GANs construct a generative model of a probability distribution $P$ by setting up an adversarial game between a generator $g$ and a discriminator $d$:

$$\min_G \max_D \mathbb{E}_{x \sim P} \log d(x) + \mathbb{E}_{z \sim Q} \left(1 - \log d(g(z))\right).$$

In practice, most of the applications of GANs concern modeling distributions of natural images. In these cases, both the generator $g$ and the discriminator $d$ are parametrized as deep convnets (LeCun et al., 1998). Among the multiple architectural variations explored in the literature, the most prominent is the Deep Convolutional Generative Adversarial Network (DCGAN) (Radford et al., 2015). Therefore, in this paper we will use the specification of the generator function of the DCGAN to construct the generator of GLO across all of our experiments.

**Autoencoders.** In their simplest form, an Auto-Encoder (AE) is a pair of neural networks, formed by an encoder $f : \mathcal{X} \to \mathcal{Z}$ and a decoder $g : \mathcal{Z} \to \mathcal{X}$. The role of an autoencoder is the compress the

PCA

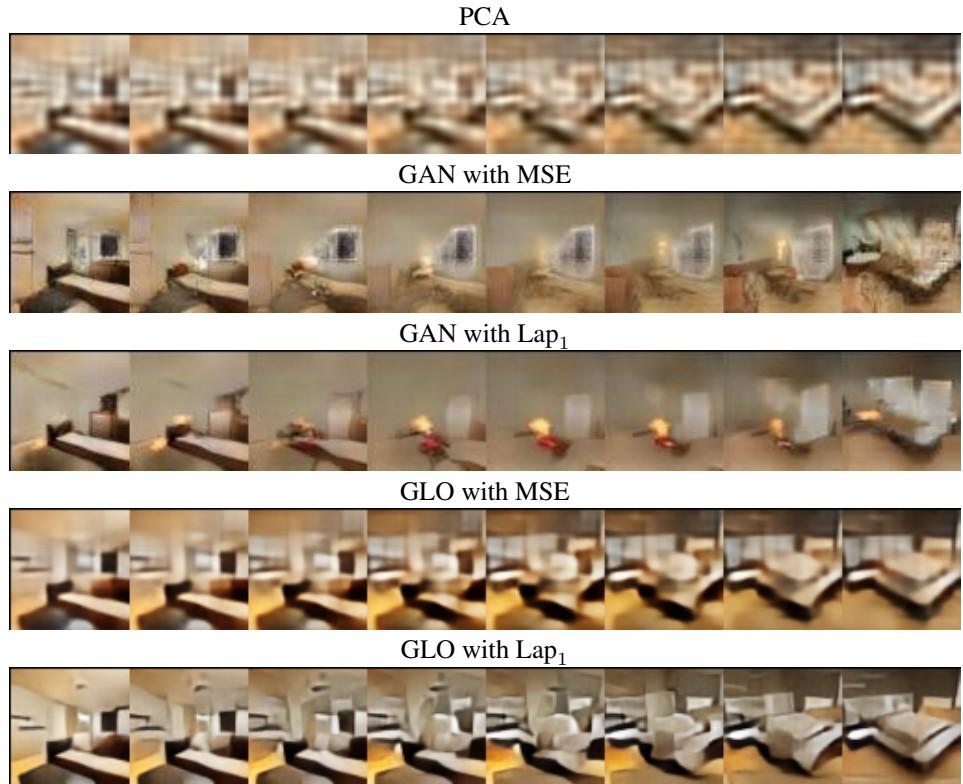

GAN with MSE

GAN with Lap$_1$

GLO with MSE

GLO with Lap$_1$

Figure 7: Interpolation of training examples on the LSUN 64×64 dataset. Both GAN and GLOs use a DCGAN generator.

data $\{x_1, \ldots, x_N\}$ into the representation $\{z_1, \ldots, z_N\}$ using the encoder $f(x_i)$, and decompress it using the decoder $g(f(x_i))$. Therefore, autoencoders minimize $\mathbb{E}_{x \sim P} \ell(g(f(x)), x)$, where $\ell : \mathcal{X} \times \mathcal{X}$ is a simple loss function, such as the mean squared error. There is a vast literature on autoencoders, spanning three decades from their conception (Bourlard and Kamp, 1988; Baldi and Hornik, 1989), renaissance (Hinton and Salakhutdinov, 2006), and recent probabilistic extensions (Vincent et al., 2008; Kingma and Welling, 2013).

Several works have combined GANs with AEs. For instance, Zhao et al. (2016) replace the discriminator of a GAN by an AE, and Ulyanov et al. (2017) replace the decoder of an AE by a generator of a GAN. Similar to GLO, these works suggest that the combination of standard pipelines can lead to good generative models. In this work we attempt one step further, to explore if learning a generator alone is possible.

**Inverting generators.** Several works attempt at recovering the latent representation of an image with respect to a generator. In particular, Lipton and Tripathi (2017); Zhu et al. (2016) show that it is possible to recover $z$ from a generated sample. Similarly, Creswell and Bharath (2016) show that it is possible to learn the inverse transformation of a generator. These works are similar to (Zeiler and Fergus, 2014), where the gradients of a particular feature of a convnet are back-propagated to the pixel space in order to visualize what that feature stands for. From a theoretical perspective, Bruna et al. (2013) explore the theoretical conditions for a network to be invertible. All of these inverting efforts are instances of the *pre-image problem*, (Kwok and Tsang, 2004).

Bora et al. (2017) have recently showed that it is possible to recover from a trained generator with compressed sensing. Similar to our work, they use a $\ell_2$ loss and backpropagate the gradient to the low rank distribution. However, they do not train the generator simultaneously. Jointly learning the representation and training the generator allows us to extend their findings. Santurkar et al. (2017) also use generative models to compress images.

1st eigenvector                    2nd eigenvector

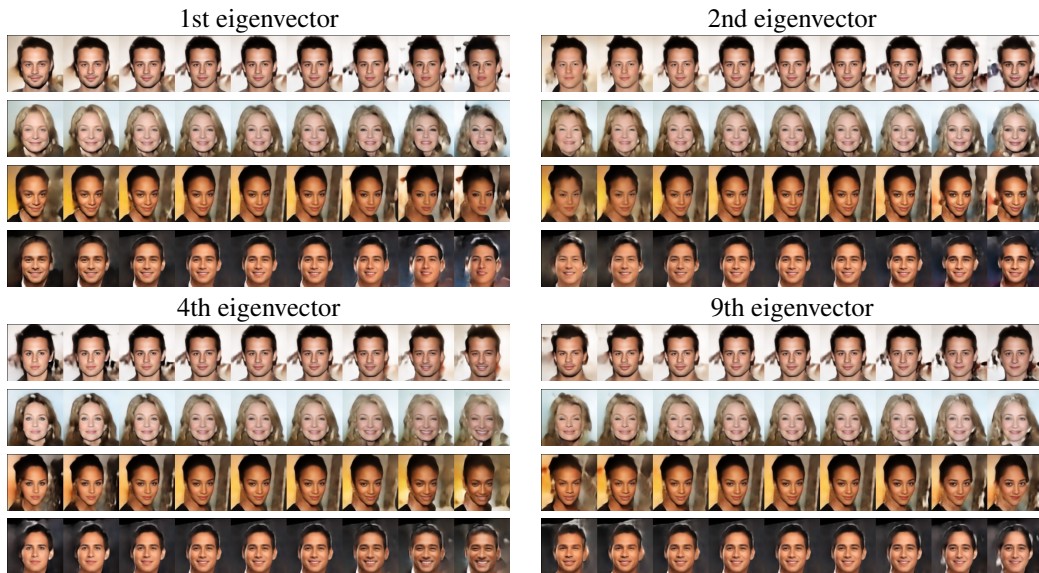

4th eigenvector                    9th eigenvector

Figure 8: Illustration of the variation around principal components of the GLO latent space on the CelebA $128 \times 128$ dataset. The original image is in the middle and we move along a eigenvector in both directions. We illustrate this process with the first 2 components as well as some later ones.

Several works have used an optimization of a latent representation for the express purpose of generating realistic images, e.g. (Portilla and Simoncelli, 2000; Nguyen et al., 2017). In these works, the total loss function optimized to generate is trained separately from the optimization of the latent representation (in the former, the loss is based on a complex wavelet transform, and in the latter, on separately trained autoencoders and classification convolutional networks). In this work we train the latent representations and the generator together from scratch; and show that at test time we may sample new latent $z$ either with simple parametric distributions or by interpolation in the latent space.

**Learning representations.** Arguably, the problem of learning representations from data in an unsupervised manner is one of the long-standing problems in machine learning (Bengio et al., 2013; LeCun et al., 2015). One of the earliest algorithms used to achieve is goal is Principal Component Analysis, or PCA (Pearson, 1901; Jolliffe). For instance, PCA has been used to learn low-dimensional representations of human faces (Turk and Pentland, 1991), or to produce a hierarchy of features (Chan et al., 2015). The nonlinear extension of PCA is an autoencoder (Baldi and Hornik, 1989), which is in turn one of the most extended algorithms to learn low-dimensional representations from data. Similar algorithms learn low-dimensional representations of data with certain structure. For instance, in sparse coding (Aharon et al., 2006; Mairal et al., 2008), the representation of one image is the linear combination of a very few elements from a dictionary of features. More recently, Zhang et al. (2016) realized the capability of deep neural networks to map large collections of images to noise vectors, and Bojanowski and Joulin (2017) exploited a similar procedure to learn visual features unsupervisedly. Similarly to us, Bojanowski and Joulin (2017) allow the noise vectors $z$ to move in order to better learn the mapping from images to noise vectors. The proposed GLO is the analogous to these works, in the opposite direction: learn a map *from* noise vectors *to* images. Finally, the idea of mapping between images and noise to learn generative models is a well known technique (Chen and Gopinath, 2000; Laparra et al., 2011; Sohl-Dickstein et al., 2015; Bordes et al., 2017).

**Nuisance Variables** One might consider the generator parameters the variables of interest, and $Z$ to be "nuisance variables". There is a classical literature on dealing with nuisance parameters while estimating the parameters of interest, including optimization methods as we have used (Stuart and Ord, 2010). In this framing, it may be better to marginalize over the nuisance variables, but for the models and data we use this is intractable.

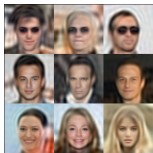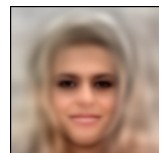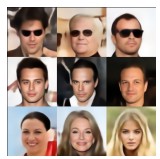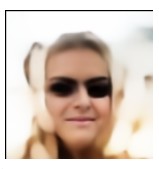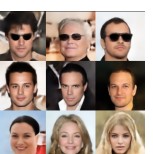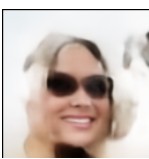

|  PCA  |  VAE with Lap$_1$  |  GLO with Lap$_1$  |

Figure 9: Illustration of feature arithmetic on CelebA. We show that by taking the average hidden representation of row **a**, substracting the one of row **b** and adding the one of row **c**, we obtain a coherent image. We show such interpolations with PCA, VAE and GLO.

**Speech generation**  Optimizing a latent representation of a generative model has a long history in speech Rabiner and Schafer (2007), both for fitting single examples in the context of fitting a generative model, and in the context of speaker adaptation.

## 5  DISCUSSION

The experimental results presented in this work suggest that, in the image domain, we can recover many of the properties of GAN models by using convnets trained with simple reconstruction losses. While this does not invalidate the promise of GANs as generic models of uncertainty or as methods for building generative models, our results suggest that, in order to more fully test the adversarial construction, research needs to move beyond images and convnets. On the other hand, practitioners who care only about generating images for a particular application, and find that the parameterized discriminator does improve their results can use reconstruction losses in their model searches, alleviating some of the instability of GAN training.

While the visual quality of the results are promising, especially on the CelebA dataset, they are not yet to the level of the results obtained by GANs on the LSUN bedrooms. This suggest several research directions: one possibility, suggested by 3, is that being able to cover the entire dataset is too onerous a task if all that is required is to generate a few nice samples. In that figure we see that GANs have trouble reconstructing randomly chosen images at the same level of fidelity as their generations. However, GANs can produce good images after a single pass through the data with SGD. In future work we hope to better understand the tension between these two observations. There are many possibilities for improving the quality of GLO samples beyond understanding the effects of coverage. For example other loss functions (e.g. a VGG metric, as in Nguyen et al. (2017)), model architectures (here we stayed close to DCGAN for ease of comparison), and more sophisticated sampling methods after training the model all may improve the visual quality of the samples.

There is also much work to be done in adding structure to the $Z$ space. Because the methods here keep track of the correspondence between samples and their representatives, and because the $Z$ space is free, we hope to be able to organize the $Z$ in interesting ways as we train.

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

Original images

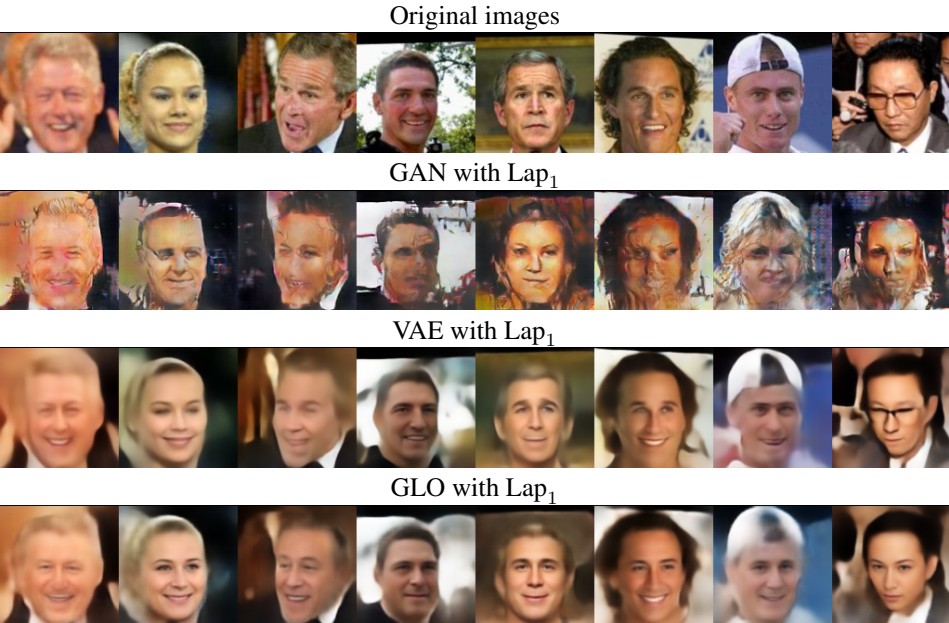

Figure 10: Reconstruction of the examples from the LFW dataset.

