# OpenReview forum: "Optimizing the Latent Space of Generative Networks"
_ICLR.cc/2018/Conference — Reject_

### Official Review · AnonReviewer1 · 2017-11-27
**OPTIMIZING THE LATENT SPACE OF GENERATIVE NETWORKS**

**Rating:** 4
**Confidence:** 4

**Review:**

Summary: The authors observe that the success of GANs can be attributed to two factors; leveraging the inductive bias of deep CNNs and the adversarial training protocol. In order to disentangle the factors of success, and they propose to eliminate the adversarial training protocol while maintaining the first factor. The proposed Generative Latent Optimization (GLO) model maps a learnable noise vector to the real images of the dataset by minimizing a reconstruction loss. The experiments are conducted on CelebA and LSUN-Bedroom datasets.

Strengths:
The paper is well written and the topic is relevant for the community.
The notations are clear, as far as I can tell, there are no technical errors.
The design choices are well motivated in Chapter 2 which makes the main idea easy to grasp.
The image reconstruction results are good.
The experiments are conducted on two challenging datasets, i.e. CelebA and LSUN-Bedroom.

Weaknesses:
A relevant model is Generative Moment Matching Network (GMMN) which can also be thought of as a “discriminator-less GAN”. However, the paper does not contrast GLO with GMMN either in the conceptual level or experimentally.

Another relevant model is Variational Autoencoders (VAE) which also learns the data distribution through a learnable latent representation by minimizing a reconstruction loss. The paper would be more convincing if it provided a comparison with VAE.

In general, having no comparisons with other models proposed in the literature as improvements over GAN such as ACGAN, InfoGAN, WGAN weakens the experimental section.

The evaluation protocol is quite weak: CelebA images are 128x128 while LSUN images are 64x64. Especially since it is a common practice nowadays to generate much higher dimensional images, i.e. 256x256, the results presented in this paper appear weak.

Although the reconstruction examples (Figure 2 and 3) are good, the image generation results (Figure 4 and 5) are worse than GAN, i.e. the 3rd images in the 2nd row in Figure 4 for instance has unrealistic artifacts, the entire Figure 5 results are quite boxy and unrealistic. The authors mention in Section 3.3.2 that they leave the careful modeling of Z to future work, however the paper is quite incomplete without this.

In Section 3.3.4, the authors claim that the latent space that GLO learns is interpretable. For example, smiling seems correlated with the hair color in Figure 6. This is a strong claim based on one example, moreover the evidence of this claim is not as obvious (based on the figure) to the reader. Moreover, in Figure 8, the authors claim that the principal components of the GLO latent space is interpretable. However it is not clear from this figure what each eigenvector generates. The authors’ observations on Figure 8 and 9 are not clearly visible through manual inspection.

Finally, as a minor note, the paper has some vague statements such as
“A linear interpolation in the noise space will generate a smooth interpolation of visually-appealing images”
“Several works attempt at recovering the latent representation of an image with respect to a generator.”
Therefore, a careful proofreading would improve the exposition.

---

### Official Review · AnonReviewer2 · 2017-11-27
**This paper is a potentially interesting alternative training procedure to GANs.**

**Rating:** 6
**Confidence:** 4

**Review:**

In this paper, the authors propose a new architecture for generative neural networks. Rather than the typical adversarial training procedure used to train a generator and a discriminator, the authors train a generator only. To ensure that noise vectors get mapped to images from the target distribution, the generator is trained to map noise vectors to the set of training images as closely as possible. Both the parameters of the generator and the noise vectors themselves are optimized during training.

Overall, I think this paper is useful. The images generated by the model are not (qualitatively and in my opinion) as high quality as extremely recent work on GANs, but do appear to be better than those produced by DCGANs. More importantly than the images produced, however, is the novel training procedure. For all of their positive attributes, the adversarial training procedure for GANs is well known to be fairly difficult to deal with. As a result, the insight that if a mapping from noise vectors to training images is learned directly, other noise images still result in natural images is interesting.

However, I do have a few questions for the authors, mostly centered around the choice of noise vectors.

In the paper, you mention that you "initialize the z by either sampling them from a Gaussian distribution or by taking the whitened PCA of the raw image pixels." What does this mean? Do you sample them from a Gaussian on some tasks, and use PCA on others? Is it fair to assume from this that the initialization of z during training matters? If so, why?

After training, you mention that you fit a full Gaussian to the noise vectors learned during training and sample from this to generate new images. I would be interested in seeing some study of the noise vectors learned during training. Are they multimodal, or is a unimodal distribution indeed sufficient? Does a Gaussian do a good job (in terms of likelihood) of fitting the noise vectors, or would some other model (even something like kernel density estimation) allow for higher probability noise vectors (and therefore potentially higher quality images) to be drawn? Does the choice of distribution even matter, or do you think uniform random vectors from the space would produce acceptable images?

---

### Official Review · AnonReviewer3 · 2017-11-28

**Rating:** 6
**Confidence:** 3

**Review:**

The paper is well written and easy to follow. I find the results very interesting. In particular the paper shows that many properties of GAN (or generative) models (e.g. interpolation, feature arithmetic) are a in great deal result of the inductive bias of deep CNN’s and can be obtained with simple reconstruction losses.

The results on CelebA seem quite remarkable for training examples (e.g. interpolation). Samples are quite good but inferior to GANs, but still impressive for the simplicity of the model. The results on SUN are a bit underwhelming, but still deliver the point reasonably well in my view. Naturally, the paper would make a much stronger claim showing good results on different datasets.

The authors mentioned that the current method can recover all the solutions that could be found by an autoencoder and reach some others. It would be very interesting to empirically explore this statement. Specifically, my intuition is that if we train a traditional autoencoder (with normalization of the latent space to match this setting) and compute the corresponding z vectors for each element in the dataset, the loss function (1) would be lower than that achieved with the proposed model. If that is true, the way of solving the problem is helping find a solution that prevents overfitting.

Following with the previous point, the authors mention that different initializations were used for the z vectors in the case of CelebA and LSUN. Does this lead to significantly different results? What would happen if the z values were initialized say with the representations learned by a fully trained deterministic autoencoder (with the normalization as in this work)? It would be good to report and discuss these alternatives in terms of loss function and results (e.g. quality of the samples).

It seems natural to include VAE baselines (using both of the losses in this work). Also, recent works have used ‘perceptual losses’, for instance for building VAE’s capable of generating sharper images:

Lamb, A., et al (2016). Discriminative regularization for generative models. arXiv preprint arXiv:1602.03220.

It would be good to compare these results with those presented in this work. One could argue that VAE’s are also mainly trained via a regularized reconstruction loss. Conversely, the proposed method can be thought as a form of autoencoder. The encoder could be thought to be implicitly defined by the optimization procedure used to recover the latent vectors in GAN's. Using explicit variables for each image would be a way of solving the optimization problem.

It would be informative to also shows reconstruction and interpolation results for a set of ‘held out’ images. Where the z values would be found as with GANs. This would test the coverage of the method and might be a way of making the comparison with GANs more relevant.

The works:

Nguyen, Anh, et al. "Synthesizing the preferred inputs for neurons in neural networks via deep generator networks." Advances in Neural Information Processing Systems. 2016.

Han, Tian, et al. "Alternating Back-Propagation for Generator Network." AAAI. 2017.

Seems very related.

---

### Author Response · Authors · 2018-01-05
**author's rebuttal 1/2**

First of all, we would like to thank the reviewers for their thoughtful comments.
The missing references suggested by reviewer R3 are indeed relevant and we will include them in the discussion of the related work in our revised draft. We would also like to thank reviewer R1 for pointing out problems with writing.

# R1: “The results on SUN are a bit underwhelming”
# R3: “the image generation results (Figure 4 and 5) are worse than GAN”

The primary focus of this work is shedding light on the success of GANs, as opposed to demonstrating a SOTA generative model for images. In particular, in this work we focused on DCGAN like architectures, without progressive generation or other more sophisticated setups.   Our aim was to understand what part of the success of DCGAN models could be explained by the inductive bias of the architecture, rather than as a result of the GAN training protocol.  We have demonstrated that on celeba, the inductive bias is key, and the GAN training protocol is not crucial, as the GAN generations are at best marginally superior to GLO generations.  On the other hand, with a DCGAN architecture,  the GAN training protocol is important on the bedrooms. We suspect that this is a capacity issue, and that the GAN “solves” the capacity issue by ignoring a large part of the training distribution, as evidenced by the reconstruction results in figure 2 and 3. Even if one does not believe this hypothesis, the discrepancy between the results on the faces and on the bedrooms is interesting as it suggests multiple other avenues for understanding the success of GANs.

In short: we fully acknowledge (here and in the text of the paper) that our generations are inferior to GAN on bedrooms and not noticeably superior to GAN on celeb; but this does not invalidate the thesis of the paper or its scientific value.


# R1: “ Especially since it is a common practice nowadays to generate much higher dimensional images, i.e. 256x256, the results presented in this paper appear weak.”

Generating 256x256 images with a DCGAN architecture on the datasets we used is still not common. In future work we will building models with more capacity and use more powerful generation protocols, with sample quality as the primary focus.


# R1: “the authors mention that different initializations were used for the z vectors in the case of CelebA and LSUN. Does this lead to significantly different results?”
# R2 “Is it fair to assume from this that the initialization of z during training matters? If so, why”

For all of the celeb images in the paper, we initialized Z with Gaussian normal vectors projected to the sphere.  For the bedrooms, we initialized with PCA.  On the faces, we found that initializing with whitened PCA leads to faster convergence, comparable reconstruction, and worse generations; and initializing with the results of an auto-encoder leads to even faster convergence, but still worse generations.   On the bedrooms, because for the models described in the paper, we are capacity limited, and because the data set is so large (and so optimization takes time), we used PCA initializations.

---

### Author Response · Authors · 2018-01-05
**author's rebuttal 2/2**


# R3 “The authors mention in Section 3.3.2 that they leave the careful modeling of Z to future work, however the paper is quite incomplete without this”

As mentioned above, our primary goal is to tease apart the influence of inductive bias via the convolutional network architecture from the GAN training protocol; as such, we have also restricted ourselves to the simplest sampling methods. Because a GAN is sampled via a simple Gaussian (or uniformly), we do the same here.   Of course we can improve sample quality (and log-likelihood) with more sophisticated model for Z, but this does not serve to help understand how a GAN is working. For example, consider figure 8 which we used to show the effects of moving principal components of the Z. We could easily make this into a method for generating new samples.

Note that DCGAN generations trained on faces will also include many “monsters” (even look at the results in the original DCGAN paper). On the other hand, we agree that on the SUN bedrooms, our model produces less convincing generations than DCGAN; but we also think the discrepancy between the results on two datasets (and indeed, the *way* in which our model's samples are less convincing paired with the way GANs fail to reconstruct training samples) is worth publishing.


# baselines (R3, R1):

We agree that VAE's are reasonable to include for comparison, and we will do so.

W.r.t. better GAN training protocols, most of the improvements have dealt with reliability. In our experiments, we trained hundreds of GAN models and picked the one with the best generations for comparison. As our goal is not to claim a SOTA image generation method, but rather to try to understand the factors in the success of a GAN, using simple techniques with with *standardized implementations* (and running them lots of times and picking the best outcomes) is preferable to getting the bleeding edge of reliable GAN training.


# GLO / AE loss (R3):

The loss from our model (with direct optimization of Z) is lower than that from an auto-encoder (although an auto-encoder fine-tuned with direct optimization of Z is comparable with our model).  However, we do agree with the intuition that random initialization serves as a kind of regularizer; we consistently see that random initialization leads to better generations.


# Reconstruction of held out images

We agree, and we will do this.

---

### Author Response · Authors · 2018-01-05
**paper revision**

We updated the draft with the following changes:
- Added VAE baselines trained on CelebA 128x128 (figures 2, 4, 6).
- Added image reconstructions of held-out images (figure 10).

---

### Decision · Program_Chairs · 2018-01-29
**ICLR 2018 Conference Acceptance Decision**

**Decision:**

Reject

**Comment:**

This paper attempts to decouple two factors underlying the success of GANs: the inductive bias of deep CNNs and adversarial training. It shows that, surprisingly, the second factor is not essential. R1 thought that comparisons to Generative Moment Matching Networks and Variational Autoencoders should be provided (note: this was added to a revised version of the paper). They also pointed out that the paper lacked comparisons to newer flavors of GANs. While R1 pointed out that the use of 128x128 and 64x64 images was weak, I tend to disagree as this is still common for many GAN papers. R2 was neutral to positive about the paper and thought that most importantly, the training procedure was novel. R3 also gave a neutral to positive review, claiming the paper was easy to follow and interesting. Like R1, R3 thought that a stronger claim could be made by using different datasets. In the rebuttal, the authors argued that the main point was not in proposing a state-of-the-art generative model of images but to provide more an introspection on the success of GANs. Overall, I found the work interesting but felt that the paper could go through one more review/revision cycle. In particular, it was very long. Without a champion, this paper did not make the cut.